# Personalized Follow Up and Genetic Diagnosis Update of *FMR1*-Related Conditions: A Change in Diagnosis, Prognosis and Expectations

**DOI:** 10.3390/ijms262010101

**Published:** 2025-10-16

**Authors:** Ana Roche-Martínez, Ariadna Ramírez-Mallafré, Lorena Joga-Elvira, Camen Manso-Bazus, Marta Rubio-Roy, Neus Baena-Diez

**Affiliations:** 1Pediatric Neurology Department, Parc Taulí University Hospital, I3PT-CERCA, UAB, 08208 Sabadell, Spain; aramirezm@tauli.cat (A.R.-M.); ljoga@tauli.cat (L.J.-E.); 2Genomic Medicine, Parc Taulí University Hospital, I3PT-CERCA, UAB, 08208 Sabadell, Spain; cmanso@tauli.cat; 3Neurology Department, Parc Taulí University Hospital, 08208 Sabadell, Spain; mrubio@tauli.cat

**Keywords:** FXS, mosaic conditions, revisited diagnosis, permutation/full mutation mosaics, preferential X chromosome inactivation

## Abstract

Fragile X syndrome (FXS, OMIM#300624) is the most common inherited cause of X-linked intellectual disability and behavior difficulties. In 99% of cases, it is caused by the pathological expansion (>200 repeats, full mutation -FM) of the CGG trinucleotide located at the 5′ UTR of the *FMR1* (*Fragile X Messenger Ribonucleoprotein 1*) gene, leading to the lack of production of the FMRP. Clinical manifestations are well known in boys but are sometimes overlooked in girls, who may remain underdiagnosed. Premutation (PM) populations (55–200 repeats) may present other medical issues, such as FXPOI or FXTAS. Mosaic conditions, such as a combination of PM and FM lines in the same patient, may lead to milder phenotypes. With the improvement of genetic testing, information regarding the exact number of CGG triplet repeats and methylation status could help explain milder phenotypes in patients who may produce some FMRP. Chromosome X preferential inactivation (XCI) in FXS women can also play a role in clinical severity. We present four non-related families who were followed up in our FXS clinic. Some of their members showed FM on Southern blot, but had milder symptoms than expected. To rule out size mosaicism, a RT-PCR was performed, giving a different and more consistent molecular diagnosis. When mosaicism was not present, methylation status was performed, excluding full methylation. For females, XCI showed preferential inactivation in one case. Revisiting old molecular diagnoses should be considered in clinical practice, especially for patients with a milder phenotype than expected from their molecular reports. This personalized follow up may change their former diagnosis, prognosis, and expectations.

## 1. Introduction

Fragile X syndrome (FXS, OMIM#300624) is the most common inherited cause of X-linked intellectual disability and behavior difficulties. In 99% of cases, it is caused by the pathological expansion (>200 repeats, full mutation -FM) of the CGG trinucleotide located at the 5′ UTR of the *FMR1 (Fragile X Messenger Ribonucleoprotein 1)* gene, which leads to a lack of production of the FMRP [1]. Clinical manifestations in boys are well described, including intellectual disability (ID), autism spectrum disorder (ASD), attention deficit and hyperactivity (ADHD), anxiety, and sleep disorders, among others. Clinical expression in girls is usually milder, leading to underdiagnosis of FXS in the female population [2]. Premutation (PM) population (55–200 repeats) may present other medical issues, such as Fragile X premature ovarian failure (FXPOI), Fragile X Tremor-Ataxia Syndrome (FXTAS) or Fragile X-Associated Neuropsychiatric Disorders (FXANDs) [3].

Mosaicism involves the presence of two or more populations of cells from a single zygote that differ in genetic constitution [4]. The *FMR1* gene presents a somatic instability and individuals may present mosaic alleles of different CGG sizes and/or methylation status within the same tissue (i.e., peripheral blood) or in tissues of a different embryological origin (peripheral blood versus saliva or bladder mucosae cells) [4,5,6].

Mosaicism in CGG repeat size and/or methylation status for PM and FM alleles may occur before germline segregation in the very early stages of development. Its prevalence has been estimated at 12% to 41% among males classified as FXS with FM [7,8,9]. Baker et al. reported that complete silencing of *FMR1* is not as frequent in FXS as previously estimated, and FXS males with a methylation mosaicism and incomplete *FMR1* silencing may be underdiagnosed [8].

Molecular diagnostic approaches for *FMR1*-associated disorders currently include Southern blot, polymerase chain reaction (PCR), and emerging long-read sequencing technologies. Southern blot has long been considered the gold standard for identifying full mutation (FM) alleles and assessing *FMR1* methylation status. However, this technique is time-consuming, requires substantial amounts of DNA, and its resolution is not sufficient to estimate CGG repeats. An accurate quantification of the CGG repeat number in premutation alleles requires the use of PCR-based techniques [10]. Conventional PCR provides a faster turnaround time but presents a limitation in amplifying GC-rich regions of *FMR1*, often missing FM alleles. Triplet-repeat-primed PCR (TP-PCR), which incorporates CGG-specific primers, improves upon conventional PCR and has become the leading method for *FMR1* testing [11,12]. It is performed with the AmplideX *FMR1* PCR/CE Kit. This assay is able to amplify alleles with more than 1000 CGG repeats and can detect both PM and FM alleles, size mosaics, and homozygous cases. Its limitations, however, are a reduced sensitivity at low DNA input concentrations (below 10 ng/μL) and a low sensitivity to low-level mosaicism [13]. Long-read sequencing technologies offer higher accuracy and the potential to overcome these limitations; however, their routine implementation in clinical settings remains limited by high costs and the complexity of downstream data analysis [14].

Improved methodologies mean we can now detect mosaic conditions that may contribute to a better understanding of the different phenotypes and a more accurate characterization of both FM and PM populations. This includes CGG number mosaicisms and methylation mosaicisms, complemented with FMRP detection or XCI in FXS women, which may explain their clinical presentation.

Information related to mosaicism is important for various reasons: it provides a more accurate molecular diagnosis for the patient and their family; it improves the clinical approach, providing a better understanding of the clinical implications of genetic conditions; and it facilitates anticipation of some medical problems. It also allows stratification among FM and PM carriers to better understand the natural history of FXS and Fragile X-Associated Conditions (FXACs).

The purpose of this work is to improve clinical management of FXS and *FMR1*-related conditions and to illustrate, with various examples, the importance of providing an appropriate molecular diagnosis. The detection of mosaicisms in FXS patients would explain milder phenotype in patients, helping to understand their difficulties and providing more accurate genetic counseling to them and their families.

## 2. Case Presentation

We present a group of non-related families who were followed up in our FXS clinic. Some of the patients, although classified as presenting FXS with full mutation detected by Southern blot, had milder symptoms than expected. An *FMR1* genetic study was repeated by our lab with updated techniques: RT-PCR was used to rule out mosaicism, methylation studies were performed to confirm full methylation or partial methylation, and XCI analysis was requested to study preferential inactivation in one young woman with FM and absence of a clear FXS phenotype. For seven members of these families, these updated molecular studies provided a different and more accurate genetic diagnosis. These findings provide a better understanding of their FXS or premutation condition, which would have otherwise remained unknown, and improve the present and future clinical approaches for these patients.

### 2.1. Family 1

Patient II.2 in Figure 1 is the mother of a high-functioning FXS teenager. She was thought to carry a full mutation (FM) without intellectual disability (high-degree studies); however, on every visit with her son, she denied anxiety or social difficulties. They explained that her brother was also an FM carrier, running a local business, with his own family, and described only mild anxiety. The genetic study (in peripheral blood) in our laboratory showed that she presents a mosaic for normal allele/premutation/full mutation (22CGG/135CGG/>200CGG). Her son (III.1) presented a PM/FM mosaic, which explains his milder phenotype and higher functionality in daily life. Her brother (II.3) was also retested, and full mutation was initially confirmed. When methylation analysis was performed (assuming the status would be “unmethylated”), a small peak of 114–150 CGG repeats (PM) was identified, together with the expected unmethylated FM allele (200CGGs).

Participant III.2, daughter of patient II.2, was approaching the age of 18 years old (yo). Despite not presenting clinical symptoms, a molecular study was performed due to the risk of early ovarian failure in case of premutation. CGG amplification was within the normal range, and she did not require further follow up. These different molecular diagnoses are summarized on Table 1.

### 2.2. Family 2

Patient II.2, pointed with the arrow in Figure 2, is a young woman in her 30s with normal neurodevelopment, and the older sister of a young FXS boy followed at the FXS Clinic. She has a high level of education and lives independently. Genetic testing was performed in another center during adulthood, with an initial molecular diagnosis of premutation. She was referred to a fertility unit for preimplantation diagnosis to prevent FXS offspring. A new genetic study performed in our lab showed a full mutation (>200 repeats) of one of the alleles, with a preferential inactivation of chromosome X (82/18), results are summarized on Table 2. The preferential inactivation of one of the X chromosomes was observed by the AR (13%:87%) and *SLITRK4* (12%:88%) loci. The *PCSK1N* locus was not found to be informative. However, it should be noted that the sample studied corresponds to one tissue (blood) and may not reflect what occurs in other tissues or other loci.

### 2.3. Family 3

Patient II.1, pointed with an arrow in Figure 3, is the first son of a healthy couple. After presenting neurodevelopmental delay, genetic testing (PCR and Southern blot) was performed in another center, revealing mosaic 45 GCC/FM, with more than 200 repeats. His mother presented PM. In the context of validation of the technique, a RT-PCR for *FMR1* was performed in a few random patients, including patient II.1 from family 3: a mosaicism for a number of triplets was found, with a combination of 45/PM/FM for patient II.1, presented in Table 3.

### 2.4. Family 4

Patients II.1 and II.2 are the only daughters of a consanguineous family, represented in Figure 4 The younger sister (II.2) presented a mild neurodevelopmental delay and was tested for FXS (Southern blot) in another hospital and identified as a carrier of normal allele of 25 CGG and a PM allele of 55CGG. In genetic consultation and as part of the updated genetic testing program mentioned above, suspecting her neurodevelopmental disorder resembled FXS, a sample of patient II.2 was reanalyzed with RT-PCR and a full mutation was detected with a mosaicism 22/55/>200 CGG. Her older sister (II.2), who had received a standard education with some support in speech and writing and presented with ADHD, was then tested, revealing a mosaicism of FM and a normal allele (25/200), data summarized on Table 4. 

## 3. Discussion

Mosaicism involves the presence of two or more populations of cells from a single zygote that differ in genetic constitution. An expanded CGG trinucleotide set in the 5′ untranslated region of the *FMR1* allele can lead to instability and size mosaicism. This is a relatively common phenomenon in *FMR1* (estimated frequency 10% to 40%), where PM and FM alleles may coexist, sometimes even with normal length alleles. In addition to size mosaicism, variation in the methylation status of full mutations can occur. Mosaicism in size or methylation has been shown to affect penetrance of the disorder and phenotype, explaining milder phenotypes.

Size mosaicism comprising both full mutation and normal-sized alleles has been described in several cases of males with FXS [15,16]. Nolin et al. [7] studied 61 FXS mosaic males and found 82% mosaic PM/FM, 3% mosaic FM/normal size allele, and 1.7% with mosaic FM/deletion. In our set of patients, two males presented PM/FM mosaic and one had normal CGG repeat size/ PM/FM mosaicism; one girl presented PM/FM mosaicism, two presented FM (new diagnostic), and one presented normal size/PM/FM.

In the first family, reviewing the mother’s molecular diagnosis allowed us to obtain an accurate diagnosis not only for her, but also for her son and brother: presenting a mosaic PM/FM modifies the risk of presenting FXTAS in adulthood, which is important information for her son and her brother, as well as for herself, who will also be at risk of presenting FXAND. This woman has not presented FXPOI, anxiety or autoimmune disorders thus far, but she will be able to identify these symptoms if they appear. Besides the size information, the methylation study of her brother confirmed the unmethylation of the FM allele (thus, there was at least some FMRP facilitating normal functioning). Although methylation cannot be used as a prognostic factor per se, it contributes to explanations of the milder phenotype.

In the second case, a young woman who thought she was permutated discovered that she presented with FM instead. With this genetic information, the patient has been able to understand better why she had some difficulties in mathematics, for example, or social anxiety when she was young. She is not expected to have premature ovarian failure (FXPOI) or tremor–ataxia (FXTAS), but her risk of having FXS offspring is higher. The preferential inactivation of her affected X chromosome helps explain her general good performance throughout her lifetime. However, the fact of her having a younger brother with special needs may have also influenced the underdiagnosis of her own mild neurodevelopmental symptoms (referred anxiety, attention difficulties, learning difficulties, etc.) during her infancy. This may open a debate about which siblings of FXS patients are completely asymptomatic, and why FXS diagnosis remains more challenging for girls.

The patient described in family 3 is an example (as is patient III.1 in the first family) of someone with a diagnosis of FXS who may develop FXTAS manifestations during adulthood in the context of PM mosaicism. If PM is identified, this information will help the clinician and the family to understand clinical evolution, paying attention to key symptoms such as tremors or ataxia when reaching adulthood [7].

The clinical history of the last family (as with the girl of the second family) represents how difficult it has been in the past to obtain an FXS diagnosis in girls. Had the younger sister not been diagnosed with FXS, the elder would have remained undiagnosed and her difficulties neglected.

We have observed (as mentioned in the literature) that patients with mosaicism for CGG repeats performed better in cognitive tests and in everyday living, with a higher autonomy compared to FM FXS patients. Given the small size of our sample, a correlation between genotype variations in mosaicism and neuropsychiatric outcomes has not been established in this study.

Updating old genetic diagnoses guided by clinically unexpected evolution can lead to a change in molecular diagnosis. Knowing the condition of PM associated with FM in the FXS populations will help clinicians to anticipate possible signs of FXTAS in adulthood, and FXPOI in girls. Offering genetic counseling to families (including mosaic information) leads to more personalized patient follow up. The identification of PM conditions facilitates setting realistic expectations for offspring.

We report a group of unrelated families who were followed up in our Fragile X clinic due to a diagnosis of FXS in at least one family member, performed during the 1990s or later, either in our center or in a different one, mainly using the Southern blot technique.

Some of the affected members of these families, although classified as having FXS, had a different evolution than expected compared to other families (milder symptoms in full mutation male or female carriers), so the genetic study was repeated by our laboratory with updated techniques (RT-PCR) and achieved a more accurate diagnosis. The two girls from family 4 would have remained undiagnosed if this genetic review had not been performed. Implications are most importantly related to personal health of these patients, but also to economic resources dedicated to the diagnosis and proper therapeutic options for these families. Other patients with different conditions of genetic origin may benefit from updating their molecular diagnoses, so they can achieve a more accurate diagnosis, prognosis, and care.

Advances in genetic diagnostic techniques and their implementation in laboratories are enabling the diagnosis of an increasing number of patients with rare diseases. For example, whole-exome sequencing (WES) allows for the detection of low-proportion mosaicism in neurodevelopmental diseases such as *PIK3CA*-related overgrowth spectrum disorders. Long-read sequencing technology has emerged as a valuable tool in discovering novel pathogenic variants in patients with unknown genetic etiologies. Furthermore, the implementation of whole-genome sequencing (WGS) allows, in a single test, for the detection of sequence variants, structural variants, methylation, detection of tandem repeat expansions, etc. Revisiting the clinical and molecular diagnosis of a patient, especially in atypical cases, should be a common strategy in clinical practice in an era where genomics is a powerful tool to better understand and support patients.

## 4. Materials and Methods

Information from patients followed up in our Fragile X clinic is regularly updated, integrating genetic, psychological, pediatric neurology, and adult neurology consultations together with our laboratory geneticist’s experience. This allows our unit to obtain a multidisciplinary overview of each case and implement coordinated actions.

Informed consent was obtained from participants before genetic retesting.

Patient genomic DNA was extracted from peripheral blood leukocytes from the patient and patient’s family. Written informed consent was obtained from the patient’s parents. The AmplideX^®^ PCR/CE FMR1 Kit (Asuragen, Inc., Austin, TX, USA), is used to determine the number of CGG repeats in the FMR1 gene using polymerase chain reaction and fragment sizing by capillary electrophoresis. The use of this kit provides accurate sizing of alleles up to 200 CGG and identification of full mutation alleles > 200 CGG, premutation alleles 55–200 CGG, and normal alleles < 55CGG. The AmplideX kit provides a characteristic product peak profile that resolves zygosity in female samples. The results were interpreted using GeneMapper™ software 6 (Thermo Fisher Scientific, Waltham, MA, USA).

Regarding the CXI pattern, a molecular study of polymorphic loci located in the *PCSK1N* (Xp11.23), *AR* (Xq12), and *SLITRK4* (Xq27.3) genes was performed to assess X chromosome inactivation, using PCR of the locus and digestion with the methylation-sensitive enzyme Hpall. Preferential inactivation values are considered to be those below 20% or above 80%, and extreme preferential inactivation values are those below 10% or above 90%. The technique used has a reliability of 99%.

A methylation study of the repetitive region (CGG)n of the *FMR1* gene was performed using Asuragen’s AmplideX FRM1 mPCR kit (Austin, TX, USA).

## 5. Conclusions

Updating old genetic diagnoses in the FXS community, guided by clinical unexpected evolution, can lead to more personalized follow ups and customized treatments, and help family planning. In some cases, it may help to detect neurological conditions that have higher risk of appearing in the future; in others, it may allow the diagnosis of not only the index case, but also of some other family members, especially girls, who may otherwise remain underdiagnosed and untreated and for whom there is a higher risk of having an offspring with a neurodevelopmental disorder. Revisiting molecular studies in undiagnosed patients should be routine for patients with neurodevelopmental disorders without a genetic diagnosis or unexpected evolution.

## Figures and Tables

**Figure 1 ijms-26-10101-f001:**
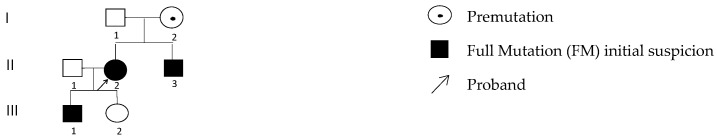
Genealogic tree of family 1, showing the proband II.2, marked with and arrow (mother, thought to be FM without symptoms, but presenting a FM/PM mosaic). I.1 is a neurotypical male, the proband’s father; I.2 is the proband’s mother, with PM; II.1 is the proband’s partner, a neurotypical male; II.3 is the proband’s brother, thought to carry a FM, but presenting a mosaic FM/PM; III.1 is the proband’s son, a FXS patient who also presented a mosaic FM/PM; III.2 is the proband’s daughter, neurotypical.

**Figure 2 ijms-26-10101-f002:**
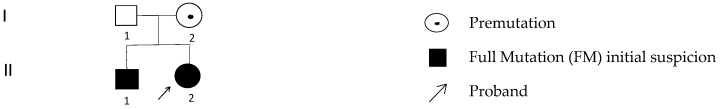
Genealogic tree of family 2. The arrow indicates the proband (II.2), a “neurotypically” developing young woman, sister of a FSX young adult (II.1), who consulted after being informed she presented a PM condition. I.1 is the father of the proband, and I.2 is the mother of the proband, carrier of a PM.

**Figure 3 ijms-26-10101-f003:**
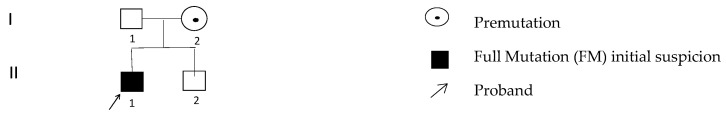
Genealogic tree of family 3. The proband, II.1, marked with an arrow, had been studied previously in another reference center using PCR and Southern blot. I.1 is the proband’s father; I.2 is the probands mother, with a PM; II.2 is the proband’s brother, with a typical neurodevelopment, and molecular testing has not been performed.

**Figure 4 ijms-26-10101-f004:**
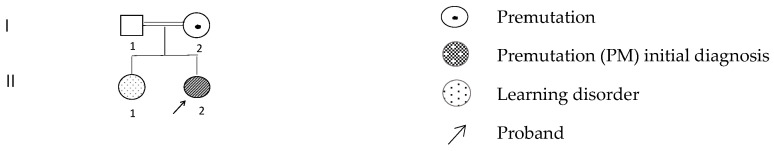
Genealogic tree of family 4. The proband (II.2) was referred to genetic consultation because of a neurodevelopmental delay of unknown origin (mosaicism 25/55CGG). Her sister (II.1) presented learning difficulties, but no genetic testing had been performed. I.1 is the father of both girls. Their mother (I.2) carries a PM.

**Table 1 ijms-26-10101-t001:** Old molecular diagnostic with Southern Blot vs. new diagnostic with RT-PRC for family 1. Participant II.3 was retested, and full mutation was initially confirmed. When methylation analysis was performed, a small peak of 114–150 CGG repeats (PM) was identified, together with the unmethylated FM allele (200CGGs).

Family	Subjects	Old Diagnostic	New Diagnostic by RT-PCR
Family 1	I.2	Premutation Southern blot	22CGG/134CGG
	II.2	Full mutation Southern blot	22CGG/135CGG/>200CGG Mosaicism N/PM/FM
	II.3	Full mutation Southern blot	Full mutation unmethylated Reanalyzed: Mosaicism PM/FM
	III.1	Full mutation Southern blot	117CGG/>200CGG Mosaicism PM/FM

**Table 2 ijms-26-10101-t002:** Old molecular diagnostic with unknown molecular technique vs. new diagnostic with RT-PRC for family 2.

Family	Subjects	Old Diagnostic	New Diagnostic by RT-PCR
Family 2	II.1	Full mutation	Full mutation
	II.2	Premutation Unknown technique	Full mutation, preferent inactivation chromosome X

**Table 3 ijms-26-10101-t003:** Old molecular diagnostic vs. new diagnostic with RT-PRC for family 3. A mosaic combining 45CGG repeats and full mutation was detected with PCR and Southern blot, then an extra 155 CGG repeat cell population was identified by RT-PCR.

Family	Subjects	Old Diagnostic	New Diagnostic with RT-PCR
Family 3	II.1	45CGG /Full mutation PCR and Southern blot	45/155/>200 CGG Mosaicism: normal/PM/FM

**Table 4 ijms-26-10101-t004:** Old molecular diagnostic vs. new diagnostic with RT-PRC for family 4. The new molecular diagnostic of FM combined with 22 and 55 CGG repeats confirmed the diagnosis of FXS for patient II.2. Her sister was subsequently tested and also diagnosed with FXS.

Family	Subjects	Old Diagnostic	New Diagnostic by RT-PCR
Family 4	II.1	Not tested	25/200CGG Full mutation
	II.2	22/55 CGG. Mosaicism Normal/PM Southern blot	22/55/200 CGG Mosaicism Normal/PM/FM

## Data Availability

The data presented in this study are available on request from the corresponding author. Clinical-specific data are restricted due to the patient’s clinical history and ethical restrictions.

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
