# Peer review of "Personalized Follow Up and Genetic Diagnosis Update of FMR1-Related Conditions: A Change in Diagnosis, Prognosis and Expectations"

_ijms, 2025, doi:10.3390/ijms262010101_

Round 1

Reviewer 1 Report

Comments and Suggestions for Authors

The merits of reassessing the genetic screening for mosaicism in FMR1 conditions are clearly conveyed in this manuscript. However, there needs to be a stronger emphasis made by the authors of why the genetic assessments were redone and why the analysis conducted suggests there should be a new clinical strategy to screen for premutations and full mutations in FMR1-related conditions. Below are my suggestions on how to do that: 

  1. The abstract should end on a summary sentence that gives the reader what do the findings of this work suggest to the broader picture of the field.
  2. There needs to be more background in the introduction that can emphasize the purpose of the study. This includes adding some of the background that is mentioned in the discussion (lines 127-143). That way the reader can be given the facts that are known before they are introduced to the unknown elements of your study. The discussion should be the author's observations of the outcome of the study rather than the background of their work. 
  3. I would also suggest adding background of what has changed in the methodology (RT-PCR). This can be added to the introduction to show what elements have been updated to be able to detect mosaicism. The discussion should also include some points that mention that improvements to genetic screening make it important for reassessments like this study for all other genetic conditions. 
  4. Other minor comments include: 
    1. A period is needed at the end of line 64. 
    2. Line 77: Who is they? The 'her' in the sentence I assume is Patient II.2
    3. The arrows in the legend should match the arrows on the the genealogic trees
    4. It may improve the visualization of the analysis if you have the genealogic trees before retesting and after retesting side by side in each of the figures for each family. That could replace using Table 1. 

Author Response

We want to thank the reviewer for all the constructive comments and suggestions to improve this paper.

English style has been reviewed, but the authors agree to send the article to the journal English Editing Service to improve the style once more, if the reviewers find it necessary.

The introduction has been rewritten and new updated references have been added to improve the quality of the text. Reference numbers have been therefore modified subsequently on the manuscript.

Complementary details have been added to “Methods” section to clarify the molecular techniques that were involved in this retrospective study.

The results have been reformulated, with shorter and more direct sentences in order to clarify the presentation of the results.

Also, the figures have been re-edited to improve the quality of the data, and footnotes explaining the genealogical trees have been rewritten.

Please find in RED the answers to your comments (in black)

Comments and Suggestions for Authors

The merits of reassessing the genetic screening for mosaicism in FMR1 conditions are clearly conveyed in this manuscript. However, there needs to be a stronger emphasis made by the authors of why the genetic assessments were redone and why the analysis conducted suggests there should be a new clinical strategy to screen for premutations and full mutations in FMR1-related conditions. Below are my suggestions on how to do that: 

Comment 1: The abstract should end on a summary sentence that gives the reader what do the findings of this work suggest to the broader picture of the field.

Response 1: This final paragraph has been added to the Abstract:

“Revisiting old molecular diagnosis should be considered in clinical practice, especially when facing patients with a milder phenotype than expected from their molecular reports. This personalized follow up can change their former diagnosis, prognosis and expectations”.

- To avoid overextension, the former final paragraph has been deleted.

“among others, a typically developed mother who thought to carry a FM, resulted PM/FM mosaic, and so was her FXS son; her typically developed brother presented an unmethylated FM; on the other hand, a typically developed young woman who thought to be PM carrier resulted FM with preferent inactivation of X chromosome (82/18)”.

Comment 2: There needs to be more background in the introduction that can emphasize the purpose of the study. This includes adding some of the background that is mentioned in the discussion (lines 127-143). That way the reader can be given the facts that are known before they are introduced to the unknown elements of your study. The discussion should be the author's observations of the outcome of the study rather than the background of their work. 

Response 2: Indeed, we agree with the reviewer that some lines of the introduction and the discussion were confusing. We have modified the text following the reviewer’s suggestions. The Introduction has been rewritten expanding the background and including some of the comments mentioned in lines 127-143, to emphasize the importance of using new techniques, applicable to the clinical follow up of SXF patients.

Please see in red, the additional material:

Fragile X syndrome (FXS, OMIM#300624) is the most common inherited cause of X-linked intellectual disability and behavior difficulties, due, in 99% of cases, to the pathological expansion (>200 repeats, full mutation -FM) of the CGG trinucleotide located at the 5' UTR of the FMR1 (Fragile X Messenger Ribonucleoprotein 1) gene, leading to the lack of production of the FMRP protein [1]. Clinical manifestations in boys are well described, with intellectual disability (ID), autism spectrum disorder (ASD), attention deficits and hyperactivity (ADHD), anxiety and sleep disorders among others; clinical expression in girls is usually milder leading to underdiagnosis of FXS in female population [2]. Premutation (PM) population (55-200 repeats) may present other medical issues, such as Fragile X premature ovarian failure (FXPOI), Fragile X Tremor-Ataxia Syndrome (FXTAS) or Fragile X Associated neuropsychiatric Disorders (FXAND) [3].

Mosaicism involves the presence of two or more populations of cells from a single zygote that differ in genetic constitution. Expanded CGG trinucleotide set in the 5’ untranslated region of the FMR1 allele can lead to instability and lead to size mosaicism [4].

The FMR1 gene presents a somatic instability and individuals may present mosaic alleles of different CGG size and/or methylation status within the same tissue (i.e. peripheral blood) or in tissues of a different embryological origin (peripheral blood versus saliva or bladder mucosae cells) [5,6]. 

Mosaicism in CGG repeat size and/or methylation status for PM and FM alleles has been estimated 12% to 41% among males who were classified as FXS with FM [7-9]. Also, Baker et al reported that complete silencing of FMR1 is not that frequent in FXS as it was previously estimated, and FXS males with a methylation mosaicism and incomplete FMR1 silencing may be underdiagnosed [7].

Molecular diagnostic approaches for FMR1-associated disorders currently include Southern blotting, polymerase chain reaction (PCR), and emerging long-read sequencing technologies. Southern blotting has long been considered the gold standard for identifying full mutation (FM) alleles and assessing FMR1 methylation status. Nevertheless, this technique is time-consuming, requires substantial amounts of, with low resolution to estimate CGG repeats; an accurate sizing of CGG repeat number in premutation alleles requires the use of PCR-based techniques [10]. Conventional PCR provides a faster turnaround time but presents a limitation in amplifying GC-rich regions of FMR1, often missing FM alleles. Triplet-repeat primed PCR (TP-PCR), which incorporates CGG-specific primers, improves upon conventional PCR and has become the leading method for FMR1 testing [11, 12], as AmplideX FMR1 PCR/CE Kit. This assay is able to amplify alleles with more than 1,000 CGG repeats and can detect both PM and FM alleles, size mosaics and homozygous cases. Its limitations, however, are a reduced sensitivity at low DNA input concentrations (below 10 ng/μL) and a low sensitivity to detect low-level mosaicism [13]. Long-read sequencing technologies offer higher accuracy and the potential to overcome these limitations; however, their routine implementation in clinical settings remains limited by high costs and the complexity of downstream data analysis [14].

Improvement in methodologies is now leading to the detection of mosaic conditions that may contribute to a better understanding of the different phenotypes and a more accurate characterization of both FM and PM populations. This includes CGG number mosaicisms, methylation mosaicisms, complemented with FMRP detection or XCI in FXS women, that may explain their clinical presentation.

Information related to mosaicism is important for various reasons: it provides a more accurate molecular diagnosis for a patient and her/his family; it improves clinical approach, providing a better understanding of the clinical implications of the genetic conditions, and facilitating anticipation of some medical problems; it also allows stratification among FM and PM carriers, to better understand FXS and Fragile X Associated Conditions (FXAC) natural history  The purpose of this work is to improve clinical management of FXS and FMR1-related conditions, illustrating with various examples the importance of providing an appropriate molecular diagnosis. The detection of mosaicisms in FXS patients would explain their milder phenotype; the reanalysis of selected permutated females would help understanding their difficulties and provide a more accurate genetic counselling to them, and their families.

Comment 3: I would also suggest adding background of what has changed in the methodology (RT-PCR). This can be added to the introduction to show what elements have been updated to be able to detect mosaicism. The discussion should also include some points that mention that improvements to genetic screening make it important for reassessments like this study for all other genetic conditions. 

Response 3: The background has been rewritten, with a review of several molecular techniques. The Discussion section has been reformulated to emphasize how the change in methodology in our lab has allowed us to detect mosaicisms, and how this may benefit other genetic conditions.

“Also, other patients with different conditions of genetic origin may benefit from updating their molecular diagnoses, so they can achieve a more accurate diagnosis, prognosis and care.

Advances in genetic diagnostic techniques and their implementation in laboratories are enabling the diagnosis of an increasing number of patients with rare diseases. For example, Whole exome sequencing (WES) allows for the detection of low-proportion mosaicism in neurodevelopmental diseases such as PIK3CA-Related Overgrowth spectrum Disorders. Long reads sequencing technology has emerged as a valuable tool in discovering novel pathogenic variants in patients with unknown genetic etiology. Furthermore, the implementation of whole genome sequencing (WGS) allows, in a single test, to detect sequence variants, structural variants, methylation, detection of tandem repeat expansions, etc. Revisiting the clinical and molecular diagnosis of a patient, especially in atypical cases, should be a common strategy in clinical practice in an era where genomics is a powerful tool to better understand and support patients”.

Other minor comments include: 

Comment 4: 1. A period is needed at the end of line 64. 

Response 4: This has been corrected.

Comment 5: 2. Line 77: Who is they? The 'her' in the sentence I assume is Patient II.2

Response 5: This has been rephrased

Comment 6: 3. The arrows in the legend should match the arrows on the the genealogic trees

Response 6: arrows in the legend have been matched to the ones on the genealogical tree. Also, the first figure has been rebuilt to improve its quality.

Comment 7: 4. It may improve the visualization of the analysis if you have the genealogic trees before retesting and after retesting side by side in each of the figures for each family. That could replace using Table 1. 

A table before and after the test has been added to the genealogic tree

The original table 1 has been deleted.

Thank you again for your comments and suggestions to improve this work.

Reviewer 2 Report

Comments and Suggestions for Authors

The manuscript presents a very interesting topic and demonstrates how updated molecular testing methods (RT-PCR, methylation analysis, etc.) can be used to update existing fragile X syndrome (FXS) diagnoses. The paper examines mosaicism effects on testing precision because this affects the diagnosis and genetic counseling for patients with untypical or mild symptoms.

The title, it is precise and informative. All references used in the manuscript are relevant, and cited correctly. All methods are vali and the variables are well defined and measured. The abstract is not clearly illustrated the aims of this manuscript. There are enough details provided to replicate the study. The conclusions answer the aim of the study.

However, there are some points in the article, that can be improved and refreshed. I find the manuscript well written in good quality of English.

The introduction section of this article is too short but the authors could provide some additional information how improved methodologies will impact the clinical and societal impact of fragile X syndrome and FMR1-related conditions.

Here are other recommendations that would improve the manuscript:

  1. The article presents clinically valuable case studies, but would be much more valuable if there were a more systematic comparison between techniques (e.g. Southern blot vs. RT-PCR). It would be good to focus on the specificity, sensitivity, and practical limitations when discussing these techniques.
  2. Are there association between genotype variations in mosaicism and neuropsychiatric outcomes?
  3. How often do the authors recommend such retesting in clinical practice?
  4. Mosaicism significantly aaffects prognosis, so the authors need to explain how they propose these data be integrated into current diagnostic and counseling?
  5. Would the results be change in different populations, and in ethnic or geographic distribution?
  6. The X chromosome inactivation patterns plays a key role in the phenotype. Can the authors explain how reliably this patterns of inactivation can be detected in clinical laboratories?Describe tissue-specific behavior and reproducibility.

The manuscript is very well written and after minor corrections, it can be accepted for publication.

Author Response

We would like to thank the reviewer for these comments and suggestions. We appreciate the quality of the review and agree with all your comments. Some of them might not be perfectly addressed due to the retrospective character of the report but we take note of them for future studies.

Please find in RED the answers to your comments (in black)

The manuscript presents a very interesting topic and demonstrates how updated molecular testing methods (RT-PCR, methylation analysis, etc.) can be used to update existing fragile X syndrome (FXS) diagnoses. The paper examines mosaicism effects on testing precision because this affects the diagnosis and genetic counseling for patients with untypical or mild symptoms.

The title, it is precise and informative. All references used in the manuscript are relevant, and cited correctly. All methods are vali and the variables are well defined and measured. The abstract is not clearly illustrated the aims of this manuscript. There are enough details provided to replicate the study. The conclusions answer the aim of the study.

However, there are some points in the article, that can be improved and refreshed. I find the manuscript well written in good quality of English.

The introduction section of this article is too short but the authors could provide some additional information how improved methodologies will impact the clinical and societal impact of fragile X syndrome and FMR1-related conditions.

Here are other recommendations that would improve the manuscript:

Comment 1: The abstract is not clearly illustrated the aims of this manuscript.

Response 1: A few clarifying sentences have been added. And a final paragraph in the abstract has been written to emphasize the aim of the study.

“Fragile X syndrome (FXS, OMIM#300624) is the most common inherited cause of X-linked intellectual disability and behavior difficulties, due, in 99% of cases, to the pathological expansion (>200 repeats, full mutation -FM) of the CGG trinucleotide located at the 5' UTR of the FMR1 (Fragile X Messenger Ribonucleoprotein 1) gene, leading to the lack of production of the FMRP protein. Clinical manifestations are well known in boys but sometimes overseen in girls who may remain underdiagnosed. Premutation (PM) population (55-200 repeats) may present other medical issues, such as FXPOI or FXTAS. Mosaic conditions as the combination of PM and FM lines in the same patient may lead to milder phenotypes. With the improvement of genetic testing, information regarding the exact number of CGG triplet repeats and methylation status could contribute to explain milder phenotypes in patients who may produce some FMRP. Also, chromosome X preferential inactivation (XCI) in FXS women can play a role in clinical severity. We present 4 non-related families who were followed up in our FXS clinic. Some of their members, with FM on Southern Blot, had milder symptoms than expected. To rule out size mosaicism, an RT-PCR was performed in our lab, proving a different and more consistent molecular diagnosis. When mosaicism was not present, methylation status was performed, excluding full methylation. For females, XCI showed preferential inactivation in one case.

Revisiting old molecular diagnosis should be considered in clinical practice, especially when facing patients with a milder phenotype than expected from their molecular reports. This personalized follow up can change their former diagnosis, prognosis and expectations”

Also, the keywords have been adjusted to better summarize the main questions addressed on the paper:

“mosaic conditions; revisited diagnosis; permutation/ full mutation mosaics; preferential X chromosome inactivation”

Comment 2: The introduction section of this article is too short but the authors could provide some additional information how improved methodologies will impact the clinical and societal impact of fragile X syndrome and FMR1-related conditions

Response 2:  We agree with the reviewer. Also, some ideas of the introduction and the discussion were confusing. We have modified the text following the reviewer’s suggestions. The Introduction has been rewritten expanding the background and including some of comments to emphasize the importance of using new techniques, applicable to the clinical follow up of FXS patients.

Coment 3: 1. The article presents clinically valuable case studies, but would be much more valuable if there were a more systematic comparison between techniques (e.g. Southern blot vs. RT-PCR). It would be good to focus on the specificity, sensitivity, and practical limitations when discussing these techniques.

Response 3: We agree with the reviewer. We have written a specific paragraph in the introduction regarding these differences between techniques, strengths and limitations:

“Conventional PCR provides a faster turnaround time but presents a limitation in amplifying GC-rich regions of FMR1, often missing FM alleles. Triplet-repeat primed PCR (TP-PCR), which incorporates CGG-specific primers, improves upon conventional PCR and has become the leading method for FMR1 testing [11, 12], as AmplideX FMR1 PCR/CE Kit. This assay is able to amplify alleles with more than 1,000 CGG repeats and can detect both PM and FM alleles, size mosaics and homozygous cases. Its limitations, however, are a reduced sensitivity at low DNA input concentrations (below 10 ng/μL) and a low sensitivity to detect low-level mosaicism [13]. Long-read sequencing technologies offer higher accuracy and the potential to overcome these limitations; however, their routine implementation in clinical settings remains limited by high costs and the complexity of downstream data analysis [14].”

Comment 4: 2. Are there association between genotype variations in mosaicism and neuropsychiatric outcomes?

Response 4: We have not been able to establish such correlation.

We have disclosed this in the final part of the discussion (lines 260-262):

“We have observed (as mentioned the literature) that patients with mosaicism for CGG repeats performed better in cognitive tests and in everyday living, with a higher autonomy than expected for FXS. Given the small size of our sample, a correlation between genotype variations in mosaicism and neuropsychiatric outcome has not been established in this study”. However, our aim is to study this phenotype differences in our broader series of 40 FXS mosaic patients in a future.

Comment 5: 3. How often do the authors recommend such retesting in clinical practice?

Response 5: We recommend considering retest if the clinical phenotype does not fit in the expected evolution. We have added this paragraph to express our opinion:

“Revisiting the clinical and molecular diagnosis of a patient, especially in atypical cases, should be a common strategy in clinical practice in an era where genomics is a powerful tool to better understand and support patients.”

Comment 6: 4. Mosaicism significantly affects prognosis, so the authors need to explain how they propose these data be integrated into current diagnostic and counseling?

Response 6: Thank you for this comment. We have added a sentence (lines 233-235) to express the purpose of integrating genetic retest in current diagnostic:

“Knowing the condition of PM associated to FM in FXS population, will help clinicians to anticipate possible signs of FXTAS in adulthood, and FXPOI in girls. Offering genetic counseling to families (including this information) improves personalized follow-ups. A change in a diagnosis of FM towards PM is a chance to customize treatments and facilitate setting realistic expectations for offspring”

Comment 7: 5. Would the results be change in different populations, and in ethnic or geographic distribution?

Response 7: This is a difficult question. In our opinion, the results can be reproduced in different populations. Different studies have been published by various groups, although generally case reports, describing FXTAS in mosaic patients with FXS. It would be very interesting to design an international cooperative study to evaluate how frequent this situation is around the world.

Comment 8: 6. The X chromosome inactivation patterns plays a key role in the phenotype. Can the authors explain how reliably this patterns of inactivation can be detected in clinical laboratories? Describe tissue-specific behavior and reproducibility.

Response 8: This is a very interesting question. We have addressed the answer both in the results, and the methods sections:

Results (family 2, section 2.2):

“Preferential inactivation of one of the X chromosomes was observed by the AR (13%:87%) and SLITRK4 (12%:88%) loci. The PCSK1N locus was not found to be informative. Preferential inactivation values are considered to be those below 20% or above 80%, and extreme preferential inactivation values are those below 10% or above 90%. The technique used has a reliability of 99%. However, it should be noted that the sample studied corresponds to one tissue (blood) and may not reflect what occurs in other tissues or other loci.

Methods:

Regarding CXI pattern, a molecular study of polymorphic loci located in the PCSK1N (Xp11.23), AR (Xq12), and SLITRK4 (Xq27.3) genes was performed to assess X chromosome inactivation, using PCR of the locus and digestion with the methylation-sensitive enzyme Hpall.

The manuscript is very well written and after minor corrections, it can be accepted for publication.

We want to thank again the reviewer for these helpful support.

Reviewer 3 Report

Comments and Suggestions for Authors

The reviewed article is devoted to genetic testing of fragile X syndrome (FXS) caused by, in most cases, pathological expansion of CGG trinucleotides. The authors indicate that the purpose of the work is to improve clinical management of FXS. They emphasized an importance of appropriate molecular diagnosis and concluded that sometimes previous genetic testing should be reviewed.

Molecular diagnostics of human genetically determined diseases are of great attention last decades. In this regard, the topic of the reviewed study is relevant.

However, the manuscript contains data on genetic testing of only a few patients from four families. This is too small cohort. It is not described how the methylation status was assessed.

The authors concluded that revisiting molecular studies in undiagnosed patients should be a routine action for patients with neurodevelopmental disorders without genetic diagnosis. This is an obvious conclusion that can be applied to any genetic disease. The work does not contain any novelty and has no scientific value.

Thus, the work appears to be of no interest. The data provided is too insignificant. There is no point in accepting the article for publication.

Author Response

We want to thank the reviewer for the constructive comments.

We have made some modifications in the abstract, introduction and methodology sections to try to improve the quality of this work.

English style has been reviewed, but the authors agree to send the article to the journal English Editing Service to improve the style once more, if the reviewers find it necessary.

Comment 1: The reviewed article is devoted to genetic testing of fragile X syndrome (FXS) caused by, in most cases, pathological expansion of CGG trinucleotides. The authors indicate that the purpose of the work is to improve clinical management of FXS. They emphasized an importance of appropriate molecular diagnosis and concluded that sometimes previous genetic testing should be reviewed.

Molecular diagnostics of human genetically determined diseases are of great attention last decades. In this regard, the topic of the reviewed study is relevant.

However, the manuscript contains data on genetic testing of only a few patients from four families. This is too small cohort. It is not described how the methylation status was assessed.

Response 1: We agree with the reviewer that this is a small cohort, and it is a retrospective study, which also limits some data access, especially regarding old genetic testing. This sample does not represent the general FXS mosaic population in our Clinic (which is around 40 in our database), but only the cases that have been reviewed due to milder phenotype presentation. In this sense, we could also argue the need of deepening into other molecular techniques, as exome sequencing, in very sever FXS cases, to rule out a second diagnosis; phenotypes may vary along the years, and HPO codification may change along the time with new clinical, neurophysiological radiological information, improving genetic approach. The intention of the authors is to create a debate on how important is to revisit both clinical phenotype and genetic information in everyday consultation, so that other pediatric neurologists and clinical geneticists will consider this possibility. This means other small cohorts in different clinics may achieve more accurate diagnosis and a better clinical management in future years. This study is mainly based on size mosaicisms, which are reported for all the patients described; regarding methylation status, this has only been studied in the adult male with FM and normal neurodevelopment, given that partial methylation was the most plausible cause of his phenotype. Methylation has not been studied for any of the females. Methylation study of the repetitive region (CGG)n of the FMR1 gene was performed using Asuragen's AmplideX FRM1 mPCR kit. We have added this last sentence to the Methods section.

Comment 2: The authors concluded that revisiting molecular studies in undiagnosed patients should be a routine action for patients with neurodevelopmental disorders without genetic diagnosis. This is an obvious conclusion that can be applied to any genetic disease. The work does not contain any novelty and has no scientific value.

Response 2: We agree with the reviewer that looking for a molecular diagnosis in undiagnosed patients is obvious; we thank you for this remark, because it might not be clear on the paper: in this case we go in a slightly different direction: we doubt of a molecular diagnosis when facing a FXS patient with a milder phenotype. We suggest the mosaic condition should be ruled out instead of accepting the patient has a milder phenotype, for premutation may involve other medical conditions during adulthood that can be anticipated. Although mosaic and methylation studies may see routinely for some FXS reference centers, other laboratories only perform Sothern Blott analyses.

Comment 3: Thus, the work appears to be of no interest. The data provided is too insignificant. There is no point in accepting the article for publication.

Response 3: Although scientific value and novelty of this paper may seem insignificant for some experts, we still think this work is worth being published, given that many laboratories provide FXS diagnosis reports without size mosaicism, and this information can be relevant for the patient.

Round 2

Reviewer 3 Report

Comments and Suggestions for Authors

The authors have significantly improved the manuscript, so it can be accepted for publication. However, there are some errors and typos that need to be corrected. Namely, Southern blotting when abbreviating a word, is written with one letter 't', i.e., must be written 'Southern blot'. In line 178, RT-PRC should be corrected to RT-PCR.

Author Response

Dear reviewer,

Thank you again for your comments.

The article is now being reviewed by the English Editing professionals of the Journal, in order to clarify wording and style.

Thank you again for your suggestions.